# Structural equation modeling to detect correlates of childhood vaccination: A moderated mediation analysis

Abraham Degarege[1]*, Karl Krupp[2,3], Vijaya Srinivas[3], Boubakari Ibrahimou[4], Purnima Madhivanan[2,3,5,6]

1 Department of Epidemiology, College of Public Health, University of Nebraska Medical Center, Omaha, Nebraska, United States of America, 2 Department of Health Promotion Sciences, Mel & Enid Zuckerman College of Public Health, The University of Arizona, Tucson, Arizona, United States of America, 3 Public Health Research Institute of India, Mysore, Karnataka, India, 4 Department of Biostatistics, Robert Stempel College of Public Health, Florida International University, Miami, Florida, United States of America, 5 Division of Infectious Diseases, College of Medicine, The University of Arizona, Tucson, Arizona, United States of America, 6 Department of Family & Community Medicine, College of Medicine, The University of Arizona, Tucson, Arizona, United States of America

* abraham.mengist@unmc.edu

## Abstract

### Objectives

This study used a health belief theory derived framework and structural equation model to examine moderators, mediators, and direct and indirect predictors of childhood vaccination.

### Methods

A secondary analysis was conducted using data collected from a cross-sectional survey of a random sample of 1599 parents living in urban and rural areas of Mysore district, India. Applying two-stage probability proportionate-to-size sampling, adolescent girls attending 7th through 10th grades in 23 schools were selected to take home a questionnaire to be answered by their parents to primarily assess HPV vaccine intentions. Parents were also asked whether their children had received one dose of Bacillus Calmette–Guérin; three doses of Diphtheria, Pertussis, Tetanus; three doses of oral Polio vaccine; and one dose of Measles vaccine. In addition, parents were asked about their attitudes towards childhood vaccination.

### Results

Out of the 1599 parents, 52.2% reported that their children had received all the routine vaccines (fully vaccinated); 42.7% reported their children had missed at least one routine vaccine, and 5.2% reported that their children had missed all routine vaccinations. Perceptions about the benefits/facilitators to childhood vaccination significantly predicted the full vaccination rate (standardized regression coefficient (β) = 0.29) directly and mediated the effect of parental education (β = 0.11) and employment (β = -0.06) on the rate of full vaccination. Parental education was significantly associated indirectly with higher rates of full vaccination

**Data Availability Statement:** All relevant data are within the manuscript.

**Funding:** This study was funded by an Investigator-Initiated Award from Merck & Co., Inc.

awarded to PM. KK was funded as a US Scholar by Fogarty International Center and National Heart Lung and Blood Institute, and National Institute of Neurological Disorders and Stroke and of the National Institutes of Health under Award Number D43 TW010540. PM was partially funded by grants from National Institutes for Health (D43-TW010540; R15-AI128714). The funder had no role in the study design, data collection, analysis, interpretation or publication of the manuscript."

**Competing interests:** This study was awarded from Merck & Co., Inc. This does not alter our adherence to PLOS ONE policies on sharing data and materials.

(β = 0.11). Parental employment was significantly associated indirectly with decreasing rates of full vaccination (β = -0.05). Area of residence moderated the role of religion (β = 0.24) and the 'number of children' in a family (β = 0.33) on parental perceptions about barriers to childhood vaccination. The model to data fit was acceptable (Root Mean Square Error of Approximation = 0.02, 95% CI 0.018 to 0.023; Comparative Fit Index = 0.92; Tucker–Lewis Index = 0.91).

## Conclusions

Full vaccination rate was relatively low among children in Mysore, especially among parents who were unsure about the benefits of routine vaccination and those with low educational levels. Interventions increasing awareness of the benefits of childhood vaccination that target rural parents with lower levels of education may help increase the rate of full childhood vaccination in India.

## Introduction

Childhood immunization remains one of the most cost-effective and widely used strategies to reduce morbidity and mortality from vaccine-preventable diseases. Annually, vaccination prevents the death of two to three million children worldwide [1]. More child deaths can be prevented by further improving vaccination levels, coverage, and timeliness [1]. In order to reduce childhood mortality, the World Health Organization (WHO) launched the Expanded Program on Immunization (EPI) in 1974 with the goal of increasing immunization worldwide [2]. In 2012, the WHO proposed ambitious goals to immunize 90% of infants with three doses of diphtheria-tetanus-pertussis (DTP3) vaccine by the year 2020 [3]. The goal remains aspirational in many developing countries [1]. As of 2018, roughly 19.8 million children did not receive routine childhood vaccines [1].

India has the world's largest number of annual births and the lowest childhood immunization rate [4]. While India has made large strides in immunization during the last three decades, immunization levels remain below the WHO goal of ≥90% coverage for DTP3 vaccines by 2020 [3]. India's Universal Vaccination Program (UIP) provides one dose of Bacillus Calmette–Guérin (BCG), three doses of DPT3, three doses of oral polio vaccine (OPV), and one dose of measles-containing vaccine (MCV) to all infants younger than one year for free [5]. Indian children receiving all four routine vaccines at recommended doses are considered fully vaccinated by WHO. Children receiving fewer than recommended doses or missing any of the childhood vaccines are categorized as 'under-vaccinated/partially vaccinated', and those not receiving any vaccinations are considered 'non-vaccinated'.

Incomplete or delayed vaccination against childhood diseases can lead to increased mortality and morbidity in children [6]. Reducing child deaths through immunization requires that adequate numbers of children receive full vaccination in a timely manner [6]. As of 2016, only about 62.0% of children aged 12–23 months were fully vaccinated in India [7]. India has a large and heterogeneous population with varied cultures, sociodemographics, religion, and education status, leading to variations in vaccination levels among regions [8–11]. The presence of under-vaccinated groups in some regions may be a source of infectious disease outbreaks in the country.

Previous studies in India have examined demand/individual-related factors affecting routine vaccination [12–26]. Most of these studies, however, assumed direct relationships between sociodemographic, environmental, psychological factors and childhood vaccination in logistic regression models [12–26]. On the other hand, some predictors of childhood vaccination may have a direct effect, and some may have an indirect effect. Furthermore, some may play a mediating or moderating role between the predictors and childhood vaccination [27, 28].

Evidence based on health belief theory (HBT) suggests that factors that affect preventive behaviors such as vaccination are complex and multifaceted, and do not always act in similar manners [27, 28]. According to HBT, individual perceptions about the benefits and barriers to behavior would directly affect the practice of behavior; the effect of sociodemographic status on behavior, on the other hand, would be indirect, influencing individual perceptions about the benefits and barriers of behavior [27, 28].

In addition, as sociodemographics vary between urban and rural residents in India, [8, 9], we hypothesize that the effect of sociodemographic factors may vary by place of residence (urban and rural). Despite the availability of health frameworks that can help us examine factors affecting childhood vaccinations in a comprehensive manner, most studies on childhood vaccination in India did not use a theoretical framework [12–26]. Moreover, some predictors of childhood vaccination can be directly observed, as are some latent variables or constructs [27, 28]. As a result, methods including standard logistic regression, factor analysis, simultaneous equation modeling and path analysis may not always be appropriate to model predictors of childhood vaccination and exploring the relationships between the predictors. Health theory-driven complex models that employ a robust analytic technique (e.g. structural equation model) are needed to better explain the nature of the relationship between sociodemographic, attitudinal and environmental factors while predicting correlates of routine childhood vaccination.

HBT has been used to explain the predictors of vaccination uptake related to influenza and HPV in different populations [29–31]. According to HBT, six factors including perceived susceptibility to disease, perceived severity of a disease, expected benefits of immunization, concerns or cost associated with immunization, strategies or information sources/media that promote immunization and confidence to adopt/accept immunization would likely affect the chance of getting a vaccine [28]. Children will likely be vaccinated if parents perceive that their children are susceptible to severe diseases, believe that the benefit their children receive through immunization (e.g., reduce susceptibility to severe diseases) outweighs the cost/concerns about the vaccine, and children appear open to receiving the vaccination [28]. HBT has, however, been developed to understand the predictors of disease prevention behaviors in the United States and is mostly used as a framework for explaining vaccination in Western countries. In addition, the theory lacks the socio-cultural factors which could affect individuals' beliefs and behaviors related to immunization. There remains limited information on the utility of the HBT to explain parental attitudes and beliefs about childhood vaccination in developing countries like India.

Knowing sociodemographic, environmental and psychosocial correlates of childhood vaccination will inform the choice of interventions to reduce parental hesitancy to childhood vaccination. The objectives of this study were to i) determine the direct and indirect predictors of childhood vaccination, ii) examine the mediating role of parental attitudes about childhood vaccination on the relationship between sociodemographic factors and childhood vaccination, iii) test the moderating role of the area of residence on the relationship between sociodemographic factors and parental attitudes about childhood vaccination; and iv) test the overall fit of an HBT derived model for analyzing childhood vaccination data.

## Materials and methods

### Study area

The study was conducted among a random sample of parents of adolescent girls residing in urban and rural areas of Mysore district, India, between February 2010 and October 2011. Mysore district is in the southern part of Karnataka, the 6[th] largest state by area and the 8[th] by population size in India. Of the 3,001,127 people living in the district (density = 450/km$^2$), 58.5% live in rural areas (1,755,714) [32].

### Study design and participants recruitment procedure

This study is a secondary analysis of data collected for a project that examined factors affecting parental intention to accept HPV vaccine for the daughters in Mysore, India [33, 34]. Assessing factors correlated with childhood vaccination was a secondary objective of the project. The Indian government has approved HPV vaccination only for adolescent girls [35, 36], the parent study; thus, targeted parents who had at least one adolescent daughter during the study period.

A cross-sectional study was conducted among a random sample of parents whose daughters aged 11 to 15 years were attending 7[th] through 10[th] grades in schools located in the urban and rural areas of Mysore District. In order to increase the chance of inclusion of parents with a variety of backgrounds while ensuring the representativeness of the samples, two-stage probability proportionate-to-size sampling was applied to select adolescent girls attending public, private and religious schools located in the district.

Study participants recruitment procedures are explained in detail elsewhere [33, 37]. In brief, 12 schools in the urban area (five government, four private and three religious) and 11 schools in the rural areas (10 government and one private) of Mysore district were selected based on probability proportionate to size sampling. A program announcement was sent home with all girls in 7[th] through 10[th] grades in the selected schools, explaining the objectives of the study and inviting eligible parents to participate. Of all the girl students in the selected age group in the select schools, 800 girls from urban schools, and 850 girls from rural schools were randomly selected and provided with a questionnaire and consent form to take home to their parents. The girls were expected to return the completed questionnaire and signed a consent form within a week. Only one parent was required to complete the questionnaire. A small proportion of parents in the urban (2.7%) and rural (2.2%) areas did not return the completed questionnaire and/or signed the consent form. IRB approval for this study was obtained from Florida International University and Public Health Research Institute of India.

### Questionnaire

The questionnaire was administered in *Kannada*, the local language, to assess factors affecting general vaccination and HPV vaccination in particular. A detailed description of the questionnaire is described elsewhere [33, 34, 37]. It included items such as "*Please indicate (to the best of your knowledge whether all your children have had the following recommended vaccinations*" to assess immunization for BCG, DPT3, OPV and MCV. Responses were recorded as four categories (0 = No, 1 = Not sure, 2 = Yes, 3 = Not applicable). Twenty items (Table 1) assessed parental attitudes about childhood vaccination, and seven items described the sociodemographic status of parents (gender, age in years, marital status, religion, occupation, educational status, number of children in a family) (Table 2). The 20 items used to examine parental attitudes about childhood vaccination were grouped into two constructs: facilitators/benefits of childhood vaccination (12 items), and barriers to childhood vaccination (8 items). Responses

**Table 1. Latent variables/constructs and the corresponding measuring items along with their responses.**

| Constructs | | Item |
|---|---|---|
| Facilitators to childhood vaccination (F) | | |
| | F1 | Vaccination are effective in preventing disease |
| | F2 | It is very important that my children receive all their vaccine |
| | F3 | Vaccine is one way that parents can ensure their child health |
| | F4 | I have a responsibility to have my children vaccinated for the protection of all children |
| | F5 | Parents should make health decision for their own children rather than a doctor |
| | F6 | The government does a good job providing vaccine and health services |
| | F7 | There are many vaccines included in the childhood vaccination schedule |
| | F8 | I would feel responsible if anything bad happened I did not have my child vaccinated |
| | F9 | I would know where to go if I wanted to have my child vaccinated |
| | F10 | When we visit a doctor/nurse, they tell us about vaccinations for my child |
| | F11 | When we visit a doctor/nurse, we ask about vaccination for our child |
| | F12 | We always get the vaccinations recommended by a doctor or nurse |
| Barriers to vaccination (B) | | |
| | B1 | I am concerned about vaccine side effects |
| | B2 | I am afraid of vaccination of my children |
| | B3 | It is better to get the disease and protected |
| | B4 | I would feel responsible if anything bad happened I had my child vaccinated |
| | B5 | The high cost of transportation would affect my decision about whether to vaccinate my child |
| | B6 | Cost is an important factor in deciding whether to vaccinate my child |
| | B7 | Getting time off from work or household duties makes it difficult to take my child for vaccination |
| | B8 | I would not give optional vaccines to my child because it is too expensive |

to items assessing parental attitudes about childhood vaccination were recorded in four categories (0 = No, 1 = Not sure, 2 = Yes, 3 = Not applicable).

## Data analysis

The M plus (version 7.3) software package was used for data analysis [38]. The main outcome variable was family childhood vaccination status grouped into three categories: fully vaccinated, under-vaccinated and no vaccination. 'Full vaccination' described families with children that had received one dose of BCG, three doses of DPT, three doses of OPV and one dose of MCV as recommended by WHO. If at least one child in a family had not received one or more of these vaccines, it was grouped as 'incomplete/under-vaccinated,' and if children had not received any vaccination, the family was grouped as 'no vaccination.'

A modified conceptual model informed by HBT was used to guide the analysis (Fig 1). Rates of full vaccination were predicted by parental perceptions about the benefits/facilitators and barriers to childhood vaccination (Table 1). Parental perceptions about the benefits/facilitators and barriers to childhood vaccination were in turn predicted by sociodemographic variables which included gender (females/males), age group (<35/≥35 years), marital status (unmarried/married), religion (Muslims/Hindus/Christians), occupation (unemployed/

**Table 2. Sociodemographic characters of the study participants/parents and vaccination status of their children at a family level in Mysore, India 2010/2011.**

| Sociodemographic Characters of the parents | Categories (n) | Vaccination status of children at a family level | | | p-value |
|---|---|---|---|---|---|
| | | Fully vaccinated (%) | Under vaccinated (%) | No vaccination (%) | |
| Education | No formal education (636) | 39.3 | 53.1 | 7.55 | <0.001 |
| | Grade 1 to 10[th] (676) | 54.0 | 41.86 | 4.1 | |
| | High school or above (287) | 76.3 | 21.3 | 2.4 | |
| Occupation | Unemployed (724) | 54.7 | 40.19 | 5.11 | 0.172 |
| | Employed (875) | 50.1 | 44.7 | 5.3 | |
| Area | Rural (822) | 42.6 | 51.3 | 6.1 | <0.001 |
| | Urban (777) | 62.3 | 33.5 | 4.3 | |
| Age (years) | ≤35 (679) | 47.3 | 46.5 | 6.2 | 0.003 |
| | >35 (920) | 55.8 | 39.8 | 4.5 | |
| Gender | Males (433) | 53.6 | 42.7 | 3.7 | 0.248 |
| | Females (1166) | 51.6 | 42.6 | 5.8 | |
| Religion | Hindus (1420) | 51.6 | 43.2 | 5.1 | 0.133 |
| | Muslims (154) | 5.1 | 40.9 | 4.6 | |
| | Christians (25) | 68.0 | 20.0 | 12.0 | |
| Marital status | Married (1487) | 52.6 | 42.2 | 5.2 | 0.441 |
| | Single (112)* | 46.4 | 48.2 | 5.4 | |
| Number of children in a family | One (131) | 55.0 | 39.7 | 5.3 | 0.773 |
| | Two (719) | 55.8 | 39.6 | 4.6 | |
| | Three (749) | 48.2 | 46.1 | 5.7 | |

Single = unmarried or separated or widowed

employed), educational status (no formal education/formal education), and the number of children in a family (one/two or three). As the number of participants reported 'not applicable' to some of the items used to assess parental attitudes about childhood vaccination were small (<20 i.e. ≤1%) or none, they were treated as missing during the analysis.

A confirmatory factor analysis was used to check the reliability and validity of the items used for measuring the constructs of perceived benefits/facilitators and perceived barriers to childhood vaccination. The strength of the influence or correlation of the scores of the items with the scores of the constructs, (i.e. the variability in the scores of the constructs explained by the items) was determined based on the magnitude of the factor loadings for each item [39]. The cut-offs used were ≤ 0.32 (poor), 0.33–0.45 (fair), 0.46–0.55 (good), 0.56–0.69 (very good), ≥ 0.70 (excellent). SEM was used to examine parental perceptions that directly predicted childhood vaccination and sociodemographic factors that indirectly predicted the behavior. SEM was also used to evaluate the mediating roles of parental perceptions on the relationship between sociodemographic factors and childhood vaccination and the moderating role of 'rural or urban residence' on the relationship between sociodemographic factors and parental perceptions about childhood vaccination. Moreover, SEM was applied to test model fit of the proposed HBT derived model (Fig 1) with the observed data.

Model fit was checked using chi-square (model acceptable if $P > 0.05$) and other fit indices including Comparative Fit Index (model acceptable if CFI > 0.90), Tucker–Lewis Index (model acceptable if TLI > 0.90), and Root Mean Square Error of Approximation (model acceptable if RMSEA is < 0.08) [40]. Before fitting the SEM model that has the measurement and structural components in it, fit of the measurement model that included the constructs 'perceived benefits/facilitators' and 'barriers' to childhood vaccination was checked. In order

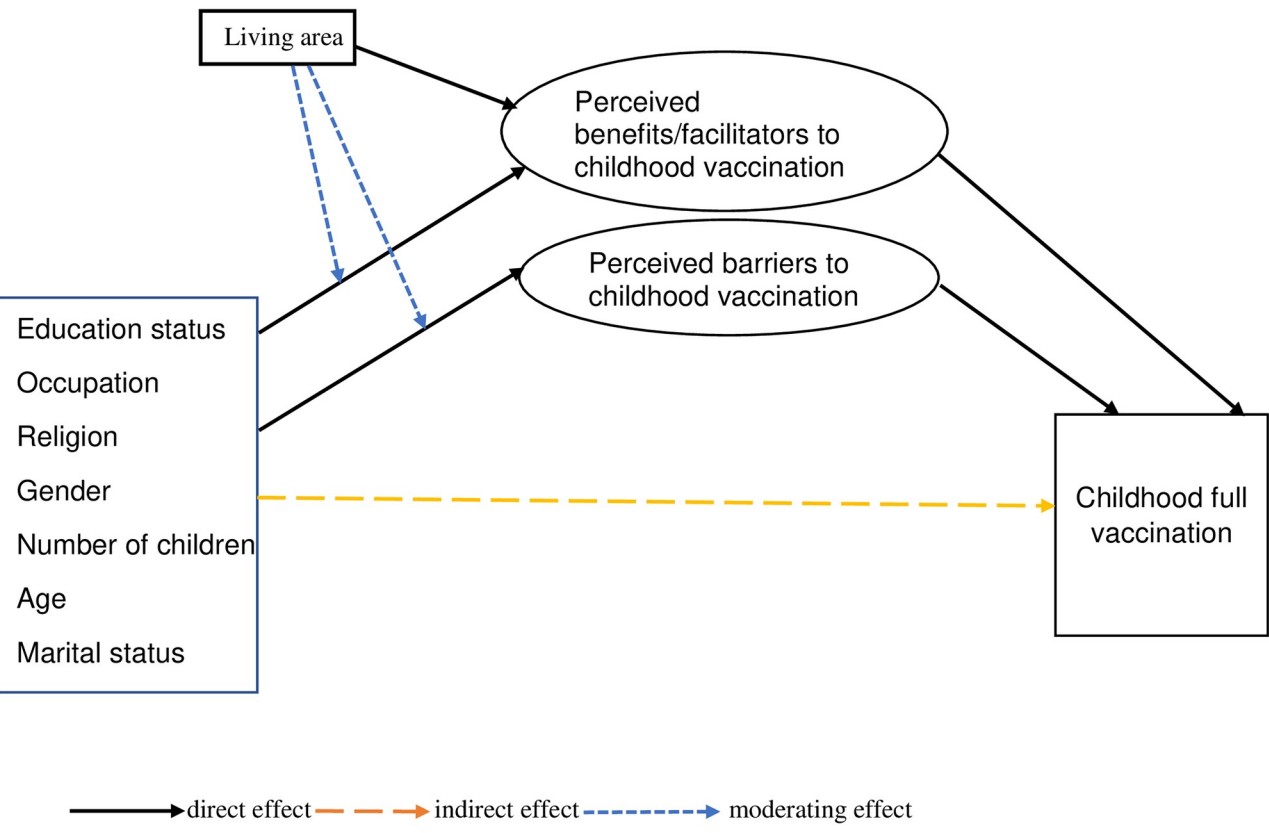

**Fig 1. Proposed health belief theory derived model for examining factors predicting full childhood vaccination.**

to improve the fit of the measurement model, a slight modification was made by removing items with factor loading < 2 and by allowing some of the items to freely covary. Similarly, we introduced slight changes in the SEM model based on the Mplus model fit indices outputs [41]. Since the outcome variable (full vaccination status) was categorical, the Weighted Least Squares Estimation Method was used to estimate the parameters (path coefficients, factor loadings, variances, and covariances) and model fit statistics [42]. Standard errors and 95% CI for direct and indirect effects were estimated using the delta method [43].

## Results

### Characteristics of the study participants

Out of 1650 parents contacted, 19 from rural and 22 from urban areas did not return the questionnaire and/or the signed the consent form. In addition, another ten parents from urban and rural areas responded to the question of uptake of routine childhood vaccines as 'not applicable' or 'not sure.' Data for the remained 1599 parents were analyzed for this study. The majority of study participants were mothers (76.7%), illiterate (63.4%), reported their religion as Hindus (99.0%), and were aged ≤35 years (53.3%). Rates of full vaccination, under vaccination and non-vaccination among children, were 52.2%, 42.7% and 5.2%, respectively. The rate of full vaccination was significantly greater among parents who were educated (60.6%), urban residents, (62.3%) and those aged >35 years (55.8%) (Table 2).

## Structural equation model

The measurement model for the construct 'perceived benefits/facilitators to childhood vaccination" that included all the items listed in Table 1 fitted acceptably to the data based on the RMSEA. But the model showed a lack of fit based on CFI and TLI values (RMSEA = 0.05, CFI = 0.89, TLI = 0.86). After allowing some residual terms for items to covary (F1 WITH F2; F3 WITH F4; F10 WITH F11; F10 WITH F12; F11 WITH F12), fit of the construct measurement model 'perceived benefits/facilitators to childhood vaccination' improved (RMSEA = 0.03, CFI = 0.97, TLI = 0.96). All 12 items listed in Table 1 were used to measure the construct 'perceived benefits/facilitators' that significantly loaded with standardized factor loadings (β) ranging from 0.27 to 0.75. The items "*It is very important that my children receive all their vaccine*" and "*I have a responsibility to have my children vaccinated for the protection of all children*" explained sufficient variance or strongly influenced/correlated (β = 0.75 and 0.73, respectively), and the item "*Parents should make health decision for their own children rather than a doctor*" explained small variance weakly influenced/correlated (β = 0.27) with the score for the construct variable 'perceived benefits/facilitator' to childhood vaccination. The remaining nine items moderately influenced/correlated with the score of the construct variable 'perceived benefits/facilitator' (β = 0.43 to 0.67).

The measurement model for the construct 'perceived barriers to childhood vaccination' fitted acceptably to the data (RMSEA = 0.06, CFI = 0.94, TLI = 0.92, β = 0.12 to 0.66), but the factor loading index for the item '*I would feel responsible, if anything bad happened I had my child vaccinated*' was 0.12. After removing items with a factor loading less than two, and allowing the residual terms for some items to covary (B5 WITH B6; B6 WITH B8), data fit of the measurement model for the construct 'barriers to childhood vaccination' improved (RMSEA = 0.05, CFI = 0.98, TLI = 0.96). Seven items, B1- B3 & B5—B8 used to measure the construct perceived barriers significantly loaded with a standardized factor loading ranging from 0.20 to 0.73. The item "*Getting time off from work or household duties makes it difficult to take my child for vaccination*" explained sufficient variance or strongly influenced/correlated (β = 0.73) and the item "*I am concerned about vaccine side effects*" explained small variance or weakly influenced/correlated (β = 0.20) with the score for the construct variable 'perceived barriers to childhood vaccination.' The observed scores for the remaining four items moderately influenced/correlated with the score of the construct variable 'perceived benefits/facilitator to childhood vaccination' (β = 0.43 to 0.67). The item "*It is better to get the disease and protected*" showed a modest-sized positive influence/correlation (β = 0.35) with the construct variable 'perceived benefits/facilitator to childhood vaccination.' The final measurement model that included 19 items used to measure facilitators and barriers to childhood vaccination had acceptable fit the data. (RMSEA = 0.04, 95% CI: 0.03–0.04; CFI = 0.93, TLI = 0.91) (Fig 2).

The final full SEM model with the structural and measurement components was identified. The total number of free parameters estimated (n = 130) was less than the number of model parameters (n = 630). The number of free parameters was the sum of the numbers of estimates, including factor loadings, variances of error and covariance. The number of model parameters is the number of unique covariances of measured variables estimated as the product of the number of measured variables (n = 35) (the number of measured variables +1) divided by 2.

Fit of the final SEM model to the data was also acceptable based on RMSEA, CFI and TLI values (RMSEA: 0.02: 95% CI: 0.018 to 0.023, CFI = 0.92; TLI = 0.91) (Fig 3). The chi-square statistic, however, suggested a significant difference in the covariance matrix in the proposed model (Fig 1) and the observed data ($\chi2$ = 3914.6, DF = 490, p < 0.01).

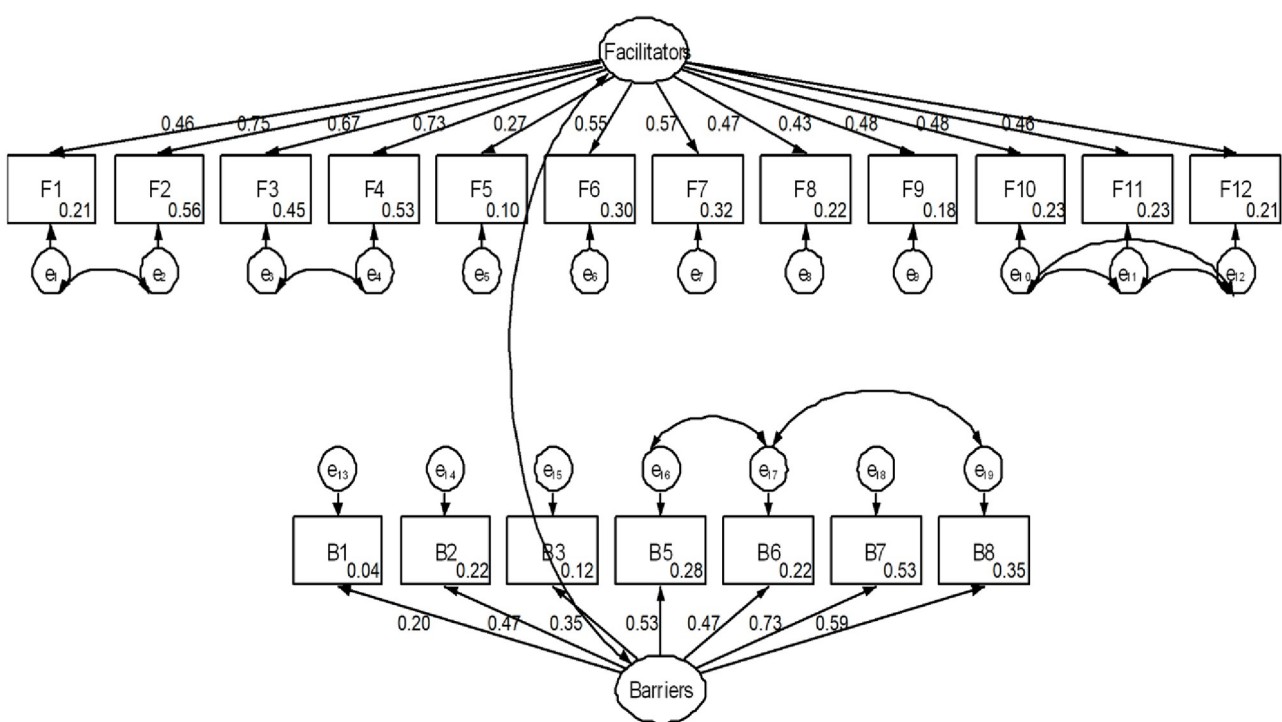

**Fig 2. Measurement model showing factor loadings of items used to measure facilitators and barriers about childhood vaccination in Mysore, India 2010/2011.**

## Factors affecting childhood vaccination status

Full childhood vaccination significantly increased with parental perceptions about the benefits/facilitators to childhood vaccination (standardized regression coefficient ($\beta$) = 0.29, $P < 0.001$). In other words, it increased if parents felt the vaccine was effective in preventing disease and ensuring child health, feeling a responsibility to protect their child and others, having a healthcare provider recommendation, receiving information about childhood vaccine from a doctor or nurse, knowing where to go for vaccinations, perceiving that the government does a good job providing vaccines and health services, and having several vaccines included in the childhood vaccination schedule. Parental education was significantly associated indirectly with increased full vaccination ($\beta$ = 0.08, $P < 0.001$). Parental employment was indirectly associated with a decreased rate of full vaccination ($\beta$ = -0.05, $P = 0.05$). The relationship of parental education ($\beta$ = 0.08, $P < 0.001$) and employment ($\beta$ = -0.06, $P = 0.045$) with the rate of full vaccination were significantly mediated by parental perceptions about the benefits/facilitators to childhood vaccination. Parental perceptions about barriers to childhood vaccination neither showed a significant association directly with the rate of full vaccination nor mediated the relationship between sociodemographic variables including age, gender, marital status, religion, education, occupation and number of children, and the rate of full vaccination (Table 3). In addition, age, gender, marital status, religion and number of children in a family had no indirect relationship with the rate of full vaccination through the construct about perceptions about the benefits/facilitators (Table 3).

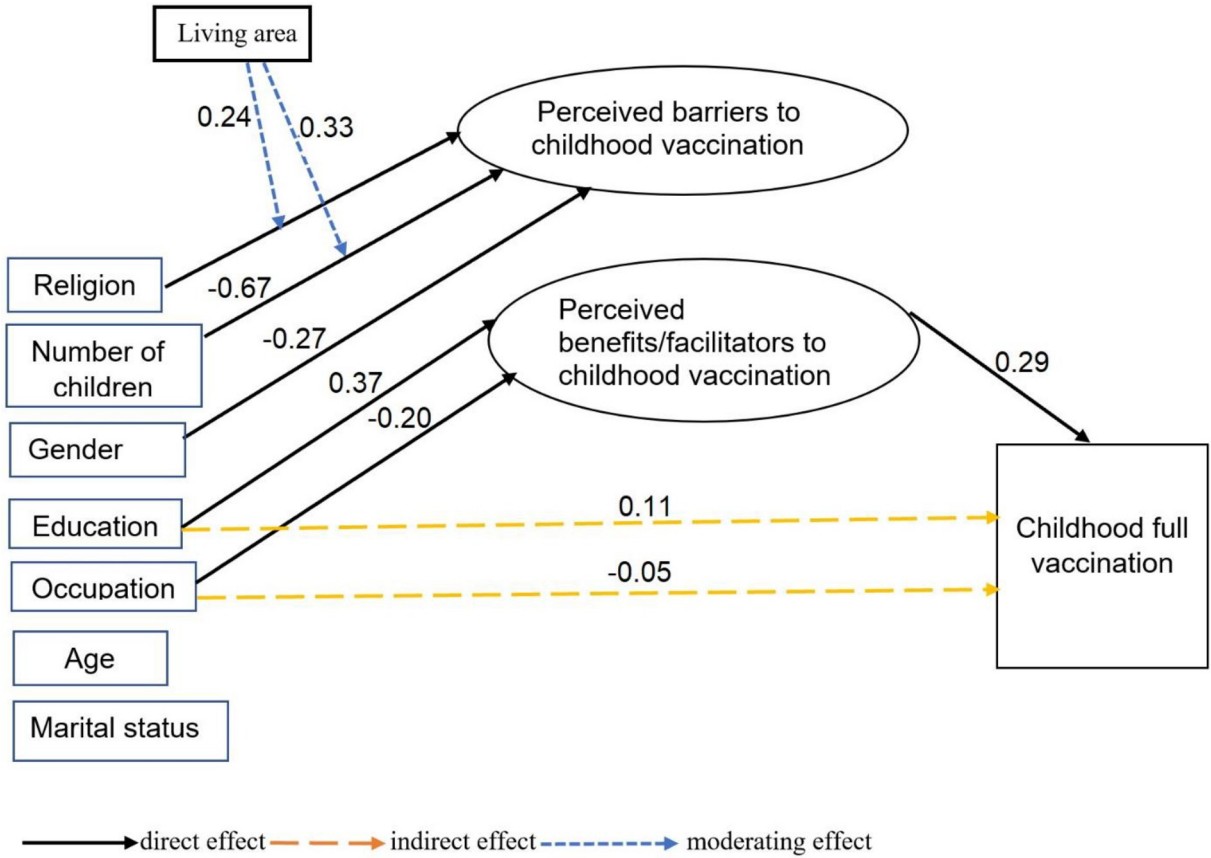

**Fig 3. Structural equation model showing moderators, mediators and direct and indirect predictors of childhood vaccination in Mysore, India, 2010/2011.**

## Effect of background factors on parental perceptions about childhood vaccination

As compared to uneducated parents, those who were educated were more likely to perceive that childhood vaccination was beneficiary ($\beta = 0.37$, $P < 0.001$). Parents living in urban areas were more likely to perceive that childhood vaccination was beneficial than those living in rural areas ($\beta = 0.96$, $P = 0.022$). Employed parents were also less likely to believe that childhood vaccination was beneficial as compared to unemployed ones ($\beta = -0.20$, $P = 0.036$). Perceptions about barriers to childhood vaccination was lower among women than in men ($\beta = -0.27$, $P = 0.030$). When compared to parents with one child, those with two or more children were less likely to perceive that childhood vaccinations were harmful ($\beta = -0.67$, $P = 0.001$). Geographical region significantly moderated the effect of religion ($\beta = 0.24$, $P = 0.045$), as did number of children ($\beta = 0.330$, $P = 0.007$) on parental perceptions about barriers to childhood vaccination. However, the nature of the relationship of parental education, gender, age and marital status with parental perceptions about benefits/facilitators and barriers to childhood vaccination did not, however, vary based on place of residence (Table 4).

## Discussion

The overall rate of full immunization among children in Mysore, India, at 52.2%, was moderate. It was significantly predicted directly by parental perceptions about benefits/facilitators to

**Table 3. Unstandardized (B) and standardized (β) effects of factors affecting full childhood vaccination for the structural equation model.**

| Factors | | | Full childhood vaccination | |
|---|---|---|---|---|
| Direct | | | *B* (95% CI) | *β* (95% CI) |
| | Facilitators | | **0.62 (0.46, 0.78)** | **0.29 (0.22, 0.36)** |
| | Barriers | | -0.12 (-0.56, 0.33) | -0.02 (-0.10, 0.06) |
| Indirect | | Mediators | | |
| | Age | Facilitators | 0.04 (-0.02, 0.09) | 0.04 (-0.02, 0.09) |
| | | Barriers | 0.002 (-0.007, 0.012) | 0.02 (-0.01, 0.01) |
| | | Sum of indirect effect | 0.04 (-0.02, 0.10) | 0.04 (-0.02, 0.10) |
| | Gender | Facilitators | 0.05 (-0.02, 0.13) | 0.05 (-0.02, 0.13) |
| | | Barriers | 0.01 (-0.02, 0.03) | 0.05 (-0.02, 0.03) |
| | | Sum of indirect effect | 0.06 (-0.02, 0.14) | 0.12 (-0.02, 0.14) |
| | Education | Facilitators | **0.11 (0.05, 0.17)** | **0.08 (0.04, 0.12)** |
| | | Barriers | 0.01 (-0.01, 0.012) | 0.002 (-0.01, 0.01) |
| | | Sum of indirect effect | **0.11 (0.05, 0.17)** | **0.08 (0.04, 0.12)** |
| | Occupation | Facilitators | **-0.06 (-0.12, -0.001)** | **-0.06 (-0.12, -0.001)** |
| | | Barriers | -0.01 (-0.02, 0.01) | -0.01 (-0.02, 0.012) |
| | | Sum of indirect effect | **-0.06 (-0.11, 0.000)** | **-0.05 (-0.11, 0.00)** |
| | Religion | Facilitators | -0.07 (-1.33, 1.20) | -0.06 (-1.32, 1.19) |
| | | Barriers | 0.01 (-0.02, 0.04) | 0.01 (-0.02, 0.04) |
| | | Sum of indirect effect | -0.06 (-1.32, 1.21) | -0.06 (-1.31, 1.20) |
| | Marital | Facilitators | 0.08 (-0.01, 0.17) | 0.078 (-0.014, 0.17) |
| | | Barriers | -0.001 (-0.01, 0.01) | -0.001(-0.01, 0.01) |
| | | Sum of indirect effect | 0.07 (-0.02, 0.17) | 0.07 (-0.015, 0.17) |

**Table 4. Standardized effects of sociodemographic factors and interaction terms on parental perceptions about benefits/facilitators and barriers to full childhood vaccination for the structural equation model.**

| Background (exposure) | Complete childhood vaccination (outcomes) | |
|---|---|---|
| | Facilitators | Barriers |
| | *β* (95% CI) | *β* (95% CI) |
| Age | 0.12 (-0.07, 0.32) | -0.12 (-0.30, 0.07) |
| Gender | 0.18 (-0.07, 0.43) | **-0.27 (-0.52, -0.03)** |
| Education | **0.37 (0.19, 0.55)** | -0.12 (-0.29, 0.05) |
| Occupation | **-0.20 (-0.39, -0.013)** | -0.05 (-0.23, 0.13) |
| Religion | -0.22 (-4.51, 4.07) | -0.37(-1.10, 0.36) |
| Marital status | 0.26 (-0.04, 0.56) | 0.07 (-0.24, 0.37) |
| Number of children | 0.21 (-0.18, 0.59) | **-0.67 (-1.05, -0.29)** |
| Area | **0.96 (0.14, 1.78)** | -0.85 (-1.63, 0.07) |
| Age* Area | 0.02 (-0.11, 0.16) | -0.03 (-0.16, 0.09) |
| Gender* Area | -0.10 (-0.26, 0.07) | 0.09 (-0.06, 0.25) |
| Education * Area | -0.11 (-0.28, 0.07) | -0.09 (-0.25, 0.08) |
| Occupation * Area | 0.08 (-0.04, 0.20) | 0.06 (-0.06, 0.17) |
| Religion * Area | -0.10 (-1.51, 1.31) | **0.24 (0.001, 0.47)** |
| Marital status * Area | -0.15 (-0.41, 0.12) | -0.19 (-0.44, 0.05) |
| Number of children*Area | -0.08 (-0.32, 0.15) | **0.33 (0.09, 0.57)** |

childhood vaccination and indirectly predicted by parental education and employment status. Place of residence moderated the effect of religion and the 'number of children' on parental perceptions about barriers to childhood vaccination.

The full childhood vaccination rate reported by urban parents was 62.3% compared to a reported rate of 42.6% in rural schools. A pooled analysis of 108,057 Indian children aged 12–23 months during a national survey for the period 2007–08 reported similar rates of full childhood vaccination of 65.6% for urban vs. 53.6% for rural residents [12]. Analysis of the National Family Health Survey data collected during 1992–93, 1998–99, 2005–06 and 2015–2016 also showed a lower vaccination rate among rural Indian children compared to those living in urban areas [9, 44, 45]. Low education and lack of awareness about the importance of childhood vaccines, poor access to healthcare and vaccines, may at least partially explain the lower rural vaccination rates [8, 9]. Indeed, when compared to urban parents (14.9%), a significantly greater portion of parents living in rural areas were illiterate (63.4%).

Parents perceiving that childhood vaccination was beneficial (e.g. effective in preventing disease, ensure their child health, help protect other children); those who felt responsible if anything bad happened did not have their child vaccinated, and those who heard about the vaccine or received a positive recommendation from a doctor or nurse were more likely to report full vaccination. In addition, parental attitudes about childhood vaccination played a mediating role between sociodemographic characteristics and the 'full vaccination' rate and this conquered with a previous report [46]. A number of studies in India, Pakistan, Nigeria, Benin and Uganda reported similar results for the relationship between levels of vaccination and parental attitudes and beliefs related to immunization [47, 48]. Similarly, a UK study showed that the need to protect children and help protect others through herd immunity was seen to influence parental decisions about vaccinations [49] positively.

These findings indicate key issues in the perceived benefits domain of HBT that could be leveraged to improve childhood vaccination rates. Community-based education campaigns focused on increasing parental awareness about the benefits of childhood vaccination in preventing disease and promoting child health may be useful. In addition, strategies led by physicians and nurses should focus on strong childhood vaccination recommendations. The psychological decision-making frameworks suggested that when individuals are uncertain, they are more open to information about vaccination in their decision-making [50]. This study also suggests that intervention to increase vaccination in children may target change in parental perception about the benefits of vaccination in protecting others, which agree with the vaccine decision-making framework [49].

Full immunization rate was also associated with the educational status of the parents. Studies in different regions of India also confirmed the correlation of full vaccination with an increased educational status of the parents [12, 17–19, 21, 24, 51, 52]. As education level increases, awareness of disease susceptibility and severity and knowledge about the importance of vaccines increases. A study in India showed that health knowledge and the ability to communicate mediated the relationship between maternal education and childhood immunization decisions [53]. It is also likely that educated parents have better access to health information, health infrastructure, and vaccination services, further facilitating decisions about childhood vaccination.

Some of the strengths of this study include having a reasonably large sample, high response rate, and probability sampling. In addition, unlike many studies in India [12–26], the current study analyzed data using a health theory-driven model with robust analytical techniques (SEM). The study also ha limitations. The chi-square test of Model Fit for the Proposed Model (Fig 1) was significant, suggesting that the covariance matrix of the proposed model and the observed data were different. The chi-square test was, however, sensitive to sample size and

approximation. Thus, the proposed model is acceptable for explaining the data. In addition, other model fit statistics, including RMSEA, CFI, and TLI values, indicated the fit of the proposed model to the current data. Moreover, mediating effects need to be interpreted in the context of a cross-sectional study. Furthermore, vaccination status for the children was reported at a family level. As there could be differential immunization status among children in a family, full immunization rates might have been falsely lowered. However, after stratifying data based on the number of children and simultaneously controlling for age, full immunization rates remained similar between parents with one child versus those with two or more. The number of children was also not associated with the full immunization rate in the SEM. It is also possible that data was influenced by recall bias; parents may not have correctly reported the vaccination status of children. Furthermore, illiterate parents might have also received support from family members or friends to respond to some questions leading to information bias.

Moreover, treating the 'not applicable' response to some items used to assess parental attitudes about childhood vaccination as missing data might have biased the estimated coefficients in the SEM model. However, the proportion of the study participants who responded as 'not applicable' were ≤1% to some items that examined parental attitudes about childhood vaccination and even none for some of them. As a result, the effect of the 'not applicable' response on the coefficient estimates could be very minimal. Finally, we are reporting on data collected from parents who had adolescent daughters attending school during the study period. As a result, the generalizability of the current results might be limited to parents with at least one school-going adolescent daughter. The generalizability of the study findings to parents without school-going adolescent girls or with only school-going boys will be further limited due to gender biases in immunization in India [54, 55]. It is also possible that parental attitudes may have changed about childhood vaccinations. Thus, policy measures designed to change negative attitudes and beliefs of parents about childhood vaccination based on the current results should be done carefully considering potential changes over the period of time. In addition, the current HBT derived framework may not be fully applicable to explain the predictors of childhood vaccination in the Indian population.

## Conclusions

In conclusion, full immunization rates were relatively low among children in Mysore, India. Misperceptions about the benefits of routine vaccination, lack of parental education and employment were barriers to receiving routine childhood vaccinations. Area of residence may modify the effect of religion and the number of children in a family on parental perceptions about barriers to childhood vaccination. Interventions are needed to improve parental awareness about the benefits of childhood vaccination, particularly among uneducated parents in rural areas. There may be other supply-related factors that require structural intervention to ensure that information and immunization are more widely available, particularly in rural areas. Future studies on supply-related factors are needed. The current study confirmed complementary relationships among sociodemographic, environment and attitudinal factors in predicting childhood vaccination based on HBT. Additional longitudinal studies are needed to confirm these relationships.

## Acknowledgments

We would like to thank the study participants for taking the time to complete the questionnaires. We would like to thank also the Block Development Officer for Mysore and the administrative staff of the schools for their assistance during data collection.

## Author Contributions

**Conceptualization:** Purnima Madhivanan.

**Data curation:** Karl Krupp, Vijaya Srinivas, Purnima Madhivanan.

**Formal analysis:** Abraham Degarege, Boubakari Ibrahimou.

**Funding acquisition:** Purnima Madhivanan.

**Investigation:** Karl Krupp, Purnima Madhivanan.

**Methodology:** Karl Krupp, Vijaya Srinivas, Purnima Madhivanan.

**Project administration:** Karl Krupp, Vijaya Srinivas, Purnima Madhivanan.

**Resources:** Karl Krupp, Purnima Madhivanan.

**Software:** Boubakari Ibrahimou.

**Supervision:** Karl Krupp, Vijaya Srinivas.

**Validation:** Karl Krupp.

**Writing – original draft:** Abraham Degarege.

**Writing – review & editing:** Abraham Degarege, Karl Krupp, Vijaya Srinivas, Boubakari Ibrahimou, Purnima Madhivanan.

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
