## [Decision Letter · Decision Letter 0]

1 Jul 2020

PONE-D-20-09473

Health Belief Theory derived framework to examine factors correlated with childhood vaccination: moderated mediation analysis using structural equation model

PLOS ONE

Dear Dr. Degarege,

Thank you for submitting your manuscript to PLOS ONE. After careful consideration, we feel that it has merit but does not fully meet PLOS ONE’s publication criteria as it currently stands. Therefore, we invite you to submit a revised version of the manuscript that addresses the points raised during the review process.

We look forward to receiving your revised manuscript.

Kind regards,

William Joe

Academic Editor

PLOS ONE

Journal Requirements:

2. Please address the following:

- Please include additional information regarding the survey or questionnaire used in the study and ensure that you have provided sufficient details that others could replicate the analyses. For instance, if you developed a questionnaire as part of this study and it is not under a copyright more restrictive than CC-BY, please include a copy, in both the original language and English, as Supporting Information.

- Please refer to any sample size calculations performed prior to participant recruitment. If these were not performed please justify the reasons. Please refer to our statistical reporting guidelines for assistance (https://journals.plos.org/plosone/s/submission-guidelines.#loc-statistical-reporting).

3.Thank you for stating the following in the Financial Disclosure section:

'This study was funded by an Investigator Initiated Award from Merck & Co., Inc.  PM received the fund ( D43 TW010540). The funders had no role in study design, data collection and analysis, decision to publish, or preparation of the manuscript.'

We note that you received funding from a commercial source: Merck & Co., Inc.

Reviewers' comments:

Reviewer's Responses to Questions

**Comments to the Author**

1. Is the manuscript technically sound, and do the data support the conclusions?

Reviewer #1: Yes

2. Has the statistical analysis been performed appropriately and rigorously? 

Reviewer #1: No

3. Have the authors made all data underlying the findings in their manuscript fully available?

Reviewer #1: Yes

4. Is the manuscript presented in an intelligible fashion and written in standard English?

Reviewer #1: No

5. Review Comments to the Author

Reviewer #1: 1. In Abstract section, method section needs to rewrite scientifically. Does the children was sample or parents of the children.

2. In the introduction section, from line number 86 to 92 “Therefore, it is imperative to……………………. immunization rates” seems the aim and objective of the study needs to be placed at the end of the introduction section.

3. As the study intended to examine health belief theory controlled and regulated on individual compositional and contextual factors, it is highly appreciated if authors can write it separately under the section “Health Belief Theory and Immunization Practices”. Line no 101 to 112. use no vaccation not unvaccinated

4. Line no 112 onwards is existing gap and support for your argument to examine HBT framework in Indian scenario followed by objectives. So, the line number 86 to 92 will also be part of this section.

5. Overall, Introduction section needed to rewrite with proper flow setting, as the line no 123-124 should not be part of this section.

6. In the Materials and Method section, Study design needs clarification. As the authors have selected adolescent girl children. In Indian scenario, gender dimension works differently. For the purpose authors are advised to go through gender biasness in immunization in India. As within demographic and health studies in developing counties including India have found disparity, discrimination and exclusion regulated by socio-economic and cultural factors contribute majorly in resource allocation and achievement which lead to poor outcome for female (child, adolescent and adult). Had the authors all these issues in mind when they were finalizing the sampling frame. So within the HBT framework, gender dimension needs to be included. Even it has not discussed in HBT framework section needs attention.

7. Further, the question is why the authors had selected female children of age 11-15 years, if they were interested in examining the regulators and predictors of basic immunization in Mysore. As per the guidelines, the basic immunization what the authors have mentioned can be completed within the age of one year as the measles vaccine is given at last after completing the age of 9 months. So, question arises that perception about vaccination may have altered over the period of 10-14 years due to various factors. So, how this HBT can be applied, is another concern.

8. Further, are all the selected schools government or private?. As the affordability to cater and send the children to these two type of school matters more. It is also important to add here that there is high possibility of very few private schools in rural areas. So, the authors have to mention the selection criteria for school is also needed.

9. Authors have mentioned about majority is living in rural areas. As per the 2011 census, close to 69% of country population and 61% of state population was living in rural areas. In case of Mysore, it is close to 58 percent. So using majority for mysore district is not appropriate here.

10. In the text author have mentioned four categories for each of the facilitators and barriers for the child vaccination but in the table option, only three categories is provided. The table can be in appendix section.

11. In the line number 142 “Only one parent in each family was required to complete the questionnaire”. It seems fine but In Indian scenario, how is it possible that 112 are unmarried in table 2 who are having girl child/children of age 11-15 years. Are they male or female?. As per the knowledge and understanding, In India, without marriage, giving birth was not acceptable. However, there may be possibilities that if the parent are unmarried, they would have adopted it or else. So, authors have to clearly mention about who are those single parent.

12. Line No 191-202 is contextualising and purpose of use of SEM to model and examine the magnitude and direction of facilitator and barrier predictors controlled on Parent SES characteristics, seems fine but need to rewrite in scientific manner. In the next section, it seems human error has happened as sign of ‘greater than’ and ‘lesser than’ is incorrect.

13. Measurement model and structural model can be single section and need methodological and scientific rigour and elaboration.

14. It is also observed that when author is using coefficient value, used “= ” sign for p values, needs correction at many places.

15. As the author used SEM to depict facilitator and barriers in child vaccination moderated by parents SES suing primary data collected from Mysore, authors should add why it is more appropriate in the study as other methods like path analysis, simultaneous equation modelling and CFA etc.

16. In the discussion section, the statement made needs clarification and updation as well. Line no-313 to line no 320 needs correction. As the srivastatava study is based on third round of DLHS survey data of India conducted during 2007-08, not at three consecutive periods of DLSH-1, DLHS-2 and DLHS-3. Further, as the author is submitting the paper in 2020, the latest available data and report for NFHS is for 2015/16 and it also provide estimates at district level for many indicators including for immunization.

17. There is an impression that study examines the HBT in child immunization behaviour using the data form mysore district, Karnataka and modelled through SEM, but in the result as well as discussion part, detailed on facilitator and barriers seems missing. Although, Parents SES are discussed with more for education.

Overall, authors are advised to restructure the paper as per framework mentioned and modelled using SEM. Sampling design written should be valid and justified. Methodology section needs linking with framework and the limitations of the same. Result and discussion section should more focus on facilitator and barriers how regulated by HBT is another concern of the paper. Authors should take care of grammatical errors and editing of the manuscript.

Thank You!!!

6. PLOS authors have the option to publish the peer review history of their article (what does this mean?). If published, this will include your full peer review and any attached files.

Reviewer #1: **Yes: **Rajesh Raushan

---

## [Author Response · Author response to Decision Letter 0]

21 Jul 2020

PONE-D-20-09473

Structural equation modeling to detect correlates of childhood vaccination: a moderated mediation analysis

PLOS ONE

Dear Dr. William Joe 

Thank you for your comments and for providing us with the reviewers' comments as well. We thank also the reviewers for their careful review and constructive comments, which have helped to improve the manuscript. We have made changes to the manuscript based on reviewers' suggestions and describe these changes in the below paragraphs. We hope that you will find our responses acceptable and we look forward to your decision.

Editors 

Response: We have checked the author guidelines for formatting of the manuscript and to assure the editor that it meets the requirements.

2. Please address the following:

- Please include additional information regarding the survey or questionnaire used in the study and ensure that you have provided sufficient details that others could replicate the analyses. For instance, if you developed a questionnaire as part of this study and it is not under a copyright more restrictive than CC-BY, please include a copy, in both the original language and English, as Supporting Information.

Response: We thank the editor for this suggestion. We have provided sufficient detail about the questionnaire used in this study that would enable others to replicate the analyses. In the manuscript, we have provided all the items and scales/response categories used to measure the outcome (childhood vaccination status) and exposure variables, including the latent (perceived benefits and perceived barriers) and measured variables (sociodemographic variables). 

- Please refer to any sample size calculations performed prior to participant recruitment. If these were not performed please justify the reasons. Please refer to our statistical reporting guidelines for assistance (https://journals.plos.org/plosone/s/submission-guidelines.#loc-statistical-reporting).

Response: This study was part of a survey implemented to examine HPV vaccine acceptance among parents of adolescent girls in India. The primary goal of the project was to examine factors associated with intention-to-vaccinate daughters with HPV vaccine. Assessing factors correlated with childhood vaccination was a secondary objective of the project. Therefore, the estimated sample size (n=1599) had enough power to test the hypotheses and answer the research question proposed related to both the objectives of the study. 

3.Thank you for stating the following in the Financial Disclosure section:

'This study was funded by an Investigator Initiated Award from Merck & Co., Inc. PM received the fund (D43 TW010540). The funders had no role in study design, data collection and analysis, decision to publish, or preparation of the manuscript.'

We note that you received funding from a commercial source: Merck & Co., Inc.

Within this Competing Interests Statement, please confirm that this does not alter your adherence to all PLOS ONE policies on sharing data and materials by including the following statement: "This does not alter our adherence to PLOS ONE policies on sharing data and materials." (as detailed online in our guide for authors http://journals.plos.org/plosone/s/competing-interests). If there are restrictions on sharing of data and/or materials, please state these. Please note that we cannot proceed with consideration of your article until this information has been declared.

Response: We have provided the text below in the manuscript under the heading “funding”

"This study was funded by an Investigator-Initiated Award from Merck & Co., Inc. awarded to PM. KK was funded as a US Scholar by Fogarty International Center and National Heart Lung and Blood Institute, and National Institute of Neurological Disorders and Stroke and of the National Institutes of Health under Award Number D43 TW010540. PM was partially funded by grants from National Institutes for Health (D43-TW010540; R15-AI128714). The funder had no role in the study design, data collection, analysis, interpretation or publication of the manuscript."

We have also included the statement which reads, "This does not alter our adherence to PLOS ONE policies on sharing data and materials." under the heading “competing interest statement” and in the cover letter. However, I didn't see the financial disclosure and competing statements in the online submission and have not updated the text there. I would greatly appreciate if you can copy this text to the online submission as needed.

Thanks

Response: We have included the amended Competing Interests Statement above in '3a' in the cover letter.

Reviewers' comments: 

Reviewer #1: 

1. In Abstract section, method section needs to rewrite scientifically. Does the children was sample or parents of the children.

Response: We have revised the method section in the abstract to clarify the sampling method. The parent study used a two stage sampling by first selecting the schools. Then a random sample of school students were selected in the age category that was needed for the study and subsequently we reached out to the parents of the selected students. We have not provided details of the sampling procedure and the questionnaire in the abstract due to word limits by the journal. The revised methods in the abstract reads as (line 26-32): 

" A cross-sectional survey was conducted among a random sample of 1599 parents living in urban and rural areas of Mysore district, India. Applying two-stage probability proportionate-to-size sampling, adolescent girls attending 7th through 10th grades in public, private and religious schools located in the district were selected to take home a questionnaire to their parents. Parents were asked whether their children had received one dose of bacillus Calmette–Guérin; three doses of Diphtheria, Pertussis, Tetanus; three doses of oral polio vaccine; and one dose of measles-containing vaccine. In addition, parents were asked about their attitudes towards childhood vaccination." 

2. In the introduction section, from line number 86 to 92 "Therefore, it is imperative to……………………. immunization rates" seems the aim and objective of the study needs to be placed at the end of the introduction section.

Response: We have rephrased the text "Therefore, it is imperative to……………………. immunization rates" and moved it to the last paragraph in the introduction before the text about the objectives of the study (line 120-122).

The revised text reads as "Knowing sociodemographic, environmental and psychosocial correlates of childhood vaccination will inform the choice of interventions to reduce parental hesitancy to childhood vaccination. The objectives of this study were …….for analyzing childhood vaccination data."

3. As the study intended to examine health belief theory controlled and regulated on individual compositional and contextual factors, it is highly appreciated if authors can write it separately under the section "Health Belief Theory and Immunization Practices". Line no 101 to 112. use no vaccination not unvaccinated.

Response: we have added a paragraph summarizing information about use of Health Belief Theory and Immunization practices. It reads as follows (line 107-119):

"HBT has been used to explain the predictors of vaccination uptake related to influenza, HPV, measles, pneumonia, and meningitis in different populations [29-31]. According to the HBT, six factors including perceived susceptibility to disease, perceived severity of a disease, expected benefits of immunization, concerns or cost associated with immunization, strategies or information sources/media that promote immunization and confidence to adopt/accept immunization would likely affect the chance of getting a vaccine [28]. Children will likely be vaccinated if parents perceive that their children are susceptible to severe diseases, believe that the benefit their children receive through immunization (e.g., reduce susceptibility to severe diseases) outweighs the cost/concerns about the vaccine, and children appear open to receiving the vaccination [28]. HBT has however, been mostly used as a framework for explaining vaccination in Western countries. In addition, the theory lacks the socio-cultural factors which could affect individuals' beliefs and behaviors related to immunization. There remains limited information on the utility of the HBT to explain parental attitudes and beliefs about childhood vaccination in developing countries like India." 

We have also replaced unvaccinated with 'no vaccination' throughout the manuscript (line 200-206 and line 195 (Table 2).

4. Line no 112 onwards is existing gap and support for your argument to examine HBT framework in Indian scenario followed by objectives. So, the line number 86 to 92 will also be part of this section.

Response: we have rephrased the text from line number 86 to 92 in the older version of the manuscript "Therefore, it is imperative to……………………. immunization rates" and moved it to the last paragraph in the introduction before the text about the objectives of the study. We used the text to support the rationale for the study.

5. Overall, Introduction section needed to rewrite with proper flow setting, as the line no 123-124 should not be part of this section.

Response: We have made major revisions in the introduction following your suggestions above and below. Please see comment 15 below for additional revisions we have made in the introduction. (line 120-122).

6. In the Materials and Method section, Study design needs clarification. As the authors have selected adolescent girl children. In Indian scenario, gender dimension works differently. For the purpose authors are advised to go through gender biasness in immunization in India. As within demographic and health studies in developing counties including India have found disparity, discrimination and exclusion regulated by socio-economic and cultural factors contribute majorly in resource allocation and achievement which lead to poor outcome for female (child, adolescent and adult). Had the authors all these issues in mind when they were finalizing the sampling frame. So within the HBT framework, gender dimension needs to be included. Even it has not discussed in HBT framework section needs attention.

Response: We have revised the study design and placed it under the heading 'Study design and participants recruitment procedure'. It reads as follows:

"A cross-sectional study was conducted among a random sample of parents whose daughters aged 11 to 15 years were attending 7th through 10th grades in schools located in the urban and rural areas of Mysore District. In order to increase the chance of inclusion of parents with a variety of backgrounds while ensuring representativeness of the samples, two-stage probability proportionate-to-size sampling was applied to select adolescent girls attending public, private and religious schools located in the district.

Study participants recruitment procedures are explained in detail elsewhere [33,34]. In brief, 12 schools in the urban area (five government, four private and three religious) and 11 schools in the rural areas (10 government and one private) of Mysore District were selected. A program announcement was sent home with all girls in 7th through 10th grades in the selected schools, explaining the objectives of the study and inviting eligible parents to participate. Of all the girl students in the selected age group in the select schools, 800 girls from urban schools, and 850 girls from rural schools were randomly selected and provided with a questionnaire and consent form to take home to their parents. The girls were expected to return the completed questionnaire and signed consent form within a week. Only one parent was required to complete the questionnaire. A small proportion of parents in the urban (2.7%) and rural (2.2%) areas did not return the completed questionnaire and/or the sign the consent form. IRB approval for this study was obtained from Florida International University and Public Health Research Institute of India."

We have also discussed the limitations of the generalizability of the current finding due to potential gender bias in immunization in India. It reads as follows (Line 428-431):

"Finally, we are reporting on data collected from parents who had adolescent daughters attending school during the study period. As a result, the generalizability of the current results might be limited to parents with at least one school-going adolescent daughter. The generalizability of the study findings to parents without school-going adolescent girls or with only school-going boys will be further limited due to gender biases in immunization in India [52,53]."

7. Further, the question is why the authors had selected female children of age 11-15 years, if they were interested in examining the regulators and predictors of basic immunization in Mysore. As per the guidelines, the basic immunization what the authors have mentioned can be completed within the age of one year as the measles vaccine is given at last after completing the age of 9 months. So, question arises that perception about vaccination may have altered over the period of 10-14 years due to various factors. So, how this HBT can be applied, is another concern.

Response: This was a secondary analysis of data from a study implemented to examine HPV vaccine acceptance of parents in India. The primary goal of the project was to examine factors affecting intention to accept HPV vaccine for the girls as the HPV vaccine is only licensed for them in India. Assessing factors correlated with childhood vaccination was a secondary objective of the project. As the Indian government approved HPV vaccination to only adolescent girls during the study period, we targeted parents who have at least one adolescent daughter during the study period. To reach out to these parents effectively, we selected adolescent girls attending schools during the study period. However, we acknowledge that since the study participant was a parent of atleast one adolescent daughter, there is a potential for selection bias and information bias as we are asking them about events that took place more than ten years ago, and this may limit the applicability of the HBT to explain factors predicting childhood vaccination in the Indian population. We have discussed this as a limitation of the study. It reads as follows (line 428-437): "Finally, we are reporting on data collected from parents who had adolescent daughters attending school during the study period. As a result, the generalizability of the current results might be limited to parents with at least one school-going adolescent daughter. The generalizability of the study findings to parents without school-going adolescent girls or with only school-going boys will be further limited due by gender biases in immunization in India [52,53]. It is also possible that there may be recall bias and parental attitudes may have changed about childhood vaccinations. Thus, policy measures designed to change negative attitudes and beliefs of parents about childhood vaccination based on the current results should be done carefully considering potential changes over the period of time."

8. Further, are all the selected schools government or private? As the affordability to cater and send the children to these two type of school matters more. It is also important to add here that there is high possibility of very few private schools in rural areas. So, the authors have to mention the selection criteria for school is also needed.

Response: In order to include parents with a wide range of background and increase the representativeness of the sample to the population in the study area, we have included parents of adolescents who were attending government, private and religious schools. The schools were selected randomly using proportionate to size. The revised text about the type of schools selected and the selection procedure reads as follows: (line 138-145)

"In order to increase the chance of inclusion of parents with a variety of backgrounds while ensuring representativeness of the samples, two-stage probability proportionate-to-size sampling was applied to select adolescent girls attending public, private and religious schools located in the district. Study participants recruitment procedures are explained in detail elsewhere [33,34]. In brief, 12 schools in the urban area (five government, four private and three religious) and 11 schools in the rural areas (10 government and one private) of Mysore district were selected based on probability proportionate to size sampling."."

9. Authors have mentioned about majority is living in rural areas. As per the 2011 census, close to 69% of country population and 61% of state population was living in rural areas. In case of Mysore, it is close to 58 percent. So using majority for Mysore district is not appropriate here.

Response: We thank the reviewer for this comment and have replaced majority with 58.5%. The revised text reads as: "Of the 3,001,127 people living in the district (density = 450/km2), 58.5% live in rural areas (1,755,714)". (line 133, 134)

10. In the text author have mentioned four categories for each of the facilitators and barriers for the child vaccination but in the table option, only three categories is provided. The table can be in appendix section.

Response: We have corrected the response categories text in the table. We have removed the column with response categories in the table and provided that same information as a footnote in the table. It reads as follows: 'Responses to all the items were recorded in four categories: 0= No, 1= Not sure, 2= Yes, 3= Not applicable." 

11. In the line number 142 "Only one parent in each family was required to complete the questionnaire". It seems fine but In Indian scenario, how is it possible that 112 are unmarried in table 2 who are having girl child/children of age 11-15 years. Are they male or female?. As per the knowledge and understanding, In India, without marriage, giving birth was not acceptable. However, there may be possibilities that if the parent are unmarried, they would have adopted it or else. So, authors have to clearly mention about who are those single parent.

Response: We thank the reviewer for pointing out this statement. We want to clarify that we have included the definition for 'single parent' in Table 2 as a footnote. It now reads as "Single=unmarried or separated or widowed". Most of the parents who reported their marital status as single were separated or widowed. 

12. Line No 191-202 is contextualizing and purpose of use of SEM to model and examine the magnitude and direction of facilitator and barrier predictors controlled on Parent SES characteristics, seems fine but need to rewrite in scientific manner. In the next section, it seems human error has happened as sign of 'greater than' and 'lesser than' is incorrect.

Response : We thank the reviewer for catching this error. We have revised the text about the SEM and have corrected the sign of 'greater than' and 'less than in the data analysis on the method section. It reads as follows (line 227-246):

"SEM was used to examine parental perceptions that directly predicted childhood vaccination, and sociodemographic factors that indirectly predicted the behavior. SEM was also used to evaluate the mediating roles of parental perceptions on the relationship between sociodemographic factors and childhood vaccination and the moderating role of 'rural or urban residence' on the relationship between sociodemographic factors and parental perceptions about childhood vaccination. Moreover, SEM was applied to test model fit of the proposed HBT derived model (Fig 1) with the observed data. 

Model fit was checked using chi-square (model acceptable if P > 0.05) and other fit indices including Comparative Fit Index (model acceptable if CFI > 0.90), Tucker–Lewis Index (model acceptable if TLI > 0.90), and Root Mean Square Error of Approximation (model acceptable if RMSEA is < 0.08) [38]. Before fitting the SEM model that has the measurement and structural components in it, fit of the measurement model that included the constructs' perceived benefits/facilitators' and 'barriers' to childhood vaccination was checked. In order to improve the fit of the measurement model, a slight modification was made by removing items with factor loading < 2 and by allowing some of the items to freely co-vary. Similarly, we introduced slight changes in the SEM model based on the Mplus model fit indices output [39]. Since the outcome variable (full vaccination status) was categorical in nature, the Weighted Least Squares Estimation Method was used to estimate the parameters (path coefficients, factor loadings, variances, and covariances) and model fit statistics [40]. Standard errors and 95% CI for direct and indirect effects were estimated using the delta method [41]."

13. Measurement model and structural model can be single section and need methodological and scientific rigor and elaboration.

Response: We have merged the text about the measurement and structural model under a single heading (Structural equation model) and revised the text to better explain the model. The revised text about the mode reads as follows (line 259-307): 

Structural Equation Model

 "The measurement model for the construct 'perceived benefits/facilitators to childhood vaccination' that included all the items listed in table 1 fitted acceptably to the data based on the RMSEA. But the model showed lack of fit based on CFI and TLI values (RMSEA = 0.05, CFI = 0.89, TLI = 0.86). After allowing some residual terms for items to covary ( F1 WITH F2; F3 WITH F4; F10 WITH F11; F10 WITH F12; F11 WITH F12), fit of the construct measurement model 'perceived benefits/facilitators to childhood vaccination' improved (RMSEA =0.03, CFI =0.97, TLI =0.96). All 12 items listed in Table 1 were used to measure the construct 'perceived benefits/facilitators' that significantly loaded with standardized factor loadings (β) ranging from 0.27 to 0.75. The items "It is very important that my children receive all their vaccine" and "I have a responsibility to have my children vaccinated for the protection of all children" explained sufficient variance or strongly influenced/correlated (β = 0.75 and 0.73, respectively), and the item "Parents should make health decision for their own children rather than a doctor" explained small variance weakly influenced/correlated (β = 0.27) with the score for the construct variable 'perceived benefits/facilitator' to childhood vaccination. The remaining nine items moderately influenced/correlated with the score of the construct variable perceived benefits/facilitator (β = 0.43 to 0.67). 

The measurement model for the construct 'perceived barriers to childhood vaccination' fitted acceptably to the data (RMSEA = 0.06, CFI = 0.94, TLI = 0.92, β = 0.12 to 0.66), but the factor loading index for the item 'I would feel responsible, if anything bad happened I had my child vaccinated' was 0.12. After removing items with a factor loading less than two, and allowing the residual terms for some items to covary (B5 WITH B6; B6 WITH B8), data fit of the measurement model for the construct 'barriers' to childhood vaccination improved (RMSEA = 0.05, CFI =0.98, TLI =0.96). Seven items, B1- B3 & B5 - B8 used to measure the construct 'perceived barriers' significantly loaded with a standardized factor loading ranging from 0.20 to 0.73. The item "Getting time off from work or household duties makes it difficult to take my child for vaccination" explained sufficient variance or strongly influenced/correlated (β = 0.73) and the item "I am concerned about vaccine side effects" explained small variance or weakly influenced/correlated (β = 0.20) with the score for the construct variable perceived barriers to childhood vaccination. The observed scores for the remaining four items moderately influenced/correlated with the score of the construct variable perceived benefits/facilitator to childhood vaccination (β = 0.43 to 0.67). The item "It is better to get the disease and protected" showed a modest-sized positive influence/correlation (β = 0.35) with the construct variable perceived benefits/facilitator to childhood vaccination. The final measurement model that included 19 items used to measure facilitators and barriers to childhood vaccination, had acceptable fit the data. (RMSEA =0.04, 95% CI: 0.03 - 0.04; CFI =0.93, TLI =0.91) (Fig 2). 

Fig 2. Measurement model: factor loadings of items used to measure facilitators and barriers about childhood vaccination in Mysore, India 2010/2011.

The final full SEM model with the structural and measurement components was identified. The total number of free parameters estimated (#130) was less than the number of model parameters (#630). The number of free parameters was the sum of the numbers of estimates, including factor loadings, variances of error and covariance. The number of model parameters is the number of unique covariances of measured variables estimated as the product of the number of measured variables (#35) (the number of measured variables +1) divided by 2. 

Fit of the final SEM model to the data was also acceptable based on RMSEA, CFI and TLI values (RMSEA: 0.02: 90% CI: 0.018 to 0.023, CFI=0.92; TLI=0.91) (Fig 3). The chi-square statistic however, suggested a significant difference in the covariance matrix in the proposed model (Fig 1) and the observed data (χ2 =3914.6, DF=490, p < 0.01).

Fig 3. Structural Equation Model showing moderators, mediators and direct and indirect predictors of childhood vaccination in Mysore, India, 2010/2011."

14. It is also observed that when author is using coefficient value, used "=" sign for p values, needs correction at many places.

Response: We have corrected use of the = sign (added space) and 'p' (capitalized, italicized and added space) in the revised manuscript.

 15. As the author used SEM to depict facilitator and barriers in child vaccination moderated by parents SES suing primary data collected from Mysore, authors should add why it is more appropriate in the study as other methods like path analysis, simultaneous equation modelling and CFA etc.

Response: We have revised the text in the introduction to give readers why SEM is a more appropriate method to model predictors of childhood vaccination and explain the relationship between the predictors effectively as compared to other methods including logistic regression analysis, path analysis, simultaneous equation modelling and CFA. The revised text reads as follows (Line 84-107):

"Most of these studies however assumed direct relationships between sociodemographic, environmental, psychological factors and childhood vaccination in logistic regression models [12-26]. On the other hand, some predictors of childhood vaccination may have a direct effect, and some may have an indirect effect. Furthermore, some may play a mediating or moderating role between the predictors and childhood vaccination [27,28]. 

Evidence based on Health Belief Theory (HBT) suggests that factors that affect preventive behaviors such as vaccination are complex and multifaceted, and do not always act in similar manners [27,28]. According to HBT, individual perceptions about the benefits and barriers to behavior would directly affect the practice of behavior, the effect of sociodemographic status on behavior on the other hand, would be indirect, influencing individual perceptions about the benefits and barriers of behavior [27,28]. In addition, as sociodemographics vary between urban and rural residents in India, [8,9], we hypothesize that the effect of sociodemographic factors may vary by place of residence (urban and rural). Despite the availability of health frameworks that can help us examine factors affecting childhood vaccinations in a comprehensive manner, most studies on childhood vaccination in India did not use a theoretical framework [12-26]. Moreover, some predictors of childhood vaccination can be directly observed, as are some latent variables or constructs [27,28]. As a result, methods including standard logistic regression, factor analysis, simultaneous equation modelling and path analysis may not always be appropriate to model predictors of childhood vaccination, and exploring the relationships between the predictors. Health theory-driven complex models that employ robust analytic techniques (e.g. Structural Equation Model) are needed to better explain the nature of the relationship between sociodemographic, attitudinal and environmental factors while predicting correlates of routine childhood vaccination."

16. In the discussion section, the statement made needs clarification and updation as well. Line no-313 to line no 320 needs correction. As the sriva statava study is based on third round of DLHS survey data of India conducted during 2007-08, not at three consecutive periods of DLSH-1, DLHS-2 and DLHS-3. Further, as the author is submitting the paper in 2020, the latest available data and report for NFHS is for 2015/16 and it also provide estimates at district level for many indicators including for immunization.

Response: We are grateful for the reviewer for finding this typo. We have revised the results of the study by Shrivastwa, et al. Vaccine. 2015;33:D99-105

The revised text in the discussion reads as follows (line 375-377): "A pooled analysis of 108,057 Indian children aged 12–23 months during a national survey for the period 2007–08 reported similar rates of full childhood vaccination of 65.6% for urban vs 53.6% for rural, residents."

We have also discussed and compared the current finding on full vaccination rate between urban vs rural residents with the Indian National Family Health Survey data collected during 2015–2016 and cited the reference. The revised text in the discussion reads as (line 377-380): "Analysis of the National Family Health Survey data collected during 1992–93, 1998–99, 2005–06 and 2015–2016 also showed a lower vaccination rate among rural Indian children compared to those living in urban areas [9, 42, 43]." 

Reference: Ministry of Health and Family Welfare. Indian fact sheet: National Family Health Survey (NFHS-4) Accessed from http://rchiips.org/NFHS/pdf/NFHS4/India.pdf in July 6 2020.

17. There is an impression that study examines the HBT in child immunization behaviour using the data from Mysore district, Karnataka and modelled through SEM, but in the result as well as discussion part, detailed on facilitator and barriers seems missing. Although, Parents SES are discussed with more for education.

Response: We have provided details on facilitators and barriers to childhood vaccination in the results and discussion based on the data in table 3 and fig 3. The revised text in the results reads as follows (line 310-328): 

"Full childhood vaccination significantly increased with parental perceptions about the benefits/facilitators to childhood vaccination (standardized regression coefficient (β) = 0.29, P < 0.001). In other words, it increased if parents felt the vaccine was effective in preventing disease and ensuring child health, feeling a responsibility to protect their child and others, having a healthcare provider recommendation, receiving information about childhood vaccine from a doctor or nurse, knowing where to go for vaccinations, perceiving that the government does a good job providing vaccines and health services, and having several vaccines included in the childhood vaccination schedule. Parental education was significantly associated indirectly with increased full vaccination (β = 0.08, P < 0.001). Parental employment was indirectly associated with a decreased rate of full vaccination (β = - 0.05, P = 0.05). The relationship of parental education (β = 0.08, P < 0.001) and employment (β = -0.06, P = 0.045) with the rate of full vaccination were significantly mediated by parental perceptions about the benefits/facilitators to childhood vaccination. Parental perceptions about barriers to childhood vaccination neither showed a significant association directly with the rate of full vaccination nor mediated the relationship between sociodemographic variables including age, gender, marital status, religion, education, occupation and number of children, and the rate of full vaccination (Table 3). In addition, age, gender, marital status, religion and number of children in a family had no indirect relationship with the rate of full vaccination through the construct about perceptions about the benefits/facilitators (Table 3)."

The revised/added text about the influence of perceived benefits/facilitators on childhood vaccination and its implications for intervention in the discussion reads as follows (line 384 to 412):

"Parents perceiving that childhood vaccination was beneficial (e.g. effective in preventing disease, ensure their child health, help protect other children); those who felt responsible if anything bad happened did not have their child vaccinated, and those who heard about the vaccine or received a positive recommendation from a doctor or nurse, were more likely to report full vaccination. In addition, parental attitudes about childhood vaccination played a mediating role between sociodemographic characteristics and 'full vaccination' rate and this conquered with a previous report [44]. A number of studies in India, Pakistan, Nigeria, Benin and Uganda reported similar results for the relationship between levels of vaccination and parental attitudes and beliefs related to immunization [45,46]. Similarly, a UK study showed that the need to protect children and help protect others through herd immunity, were seen to positively influence parental decisions about vaccinations [47].

These findings indicate key issues in the perceived benefits domain of HBT, that could be leveraged to improve childhood vaccination rates. Community-based education campaigns focused on increasing parental awareness about the benefits of childhood vaccination in preventing disease and promoting child health may be useful. In addition, strategies led by physicians and nurses should focus on strong childhood vaccination recommendations. The psychological decision-making frameworks suggested that when individuals are uncertain, they are more open to information about vaccination in their decision-making [48]. This study also suggests that intervention to increase vaccination in children may target change in parental perception about the benefits of vaccination in protecting others, which agree with the vaccine decision-making framework [47]."

Overall, authors are advised to restructure the paper as per framework mentioned and modeled using SEM. Sampling design written should be valid and justified. Methodology section needs linking with framework and the limitations of the same. Result and discussion section should more focus on facilitator and barriers how regulated by HBT is another concern of the paper. Authors should take care of grammatical errors and editing of the manuscript.

Response: We have followed your suggestions above and made the necessary changes in the abstract, introduction, methods, results and discussion in this revised version of the manuscripts. Hopefully, these changes will address your concerns. 

We have also further edited the manuscript for grammar and language.

---

## [Decision Letter · Decision Letter 1]

19 Aug 2020

PONE-D-20-09473R1

Structural equation modeling to detect correlates of childhood vaccination: a moderated mediation analysis

PLOS ONE

Dear Dr. Degarege,

Thank you for submitting your manuscript to PLOS ONE. After careful consideration, we feel that it has merit but does not fully meet PLOS ONE’s publication criteria as it currently stands. Therefore, we invite you to submit a revised version of the manuscript that addresses the points raised during the review process.

We look forward to receiving your revised manuscript.

Kind regards,

William Joe

Academic Editor

PLOS ONE

Reviewers' comments:

Reviewer's Responses to Questions

**Comments to the Author**

1. If the authors have adequately addressed your comments raised in a previous round of review and you feel that this manuscript is now acceptable for publication, you may indicate that here to bypass the “Comments to the Author” section, enter your conflict of interest statement in the “Confidential to Editor” section, and submit your "Accept" recommendation.

Reviewer #1: All comments have been addressed

Reviewer #2: (No Response)

2. Is the manuscript technically sound, and do the data support the conclusions?

Reviewer #1: Yes

Reviewer #2: Yes

3. Has the statistical analysis been performed appropriately and rigorously? 

Reviewer #1: Yes

Reviewer #2: Yes

4. Have the authors made all data underlying the findings in their manuscript fully available?

Reviewer #1: Yes

Reviewer #2: Yes

5. Is the manuscript presented in an intelligible fashion and written in standard English?

Reviewer #1: Yes

Reviewer #2: Yes

6. Review Comments to the Author

Reviewer #1: The authors have taken care of all the comments and suggestions. The manuscript seems improved and revised in logical manner. So, there is not more suggestions other than about the title of the paper.

Reviewer #2: Please refer to the attached manuscript marked up with my feedback

7. PLOS authors have the option to publish the peer review history of their article (what does this mean?). If published, this will include your full peer review and any attached files.

Reviewer #1: **Yes: **Rajesh Raushan

Reviewer #2: No

---

## [Author Response · Author response to Decision Letter 1]

22 Aug 2020

Dear Dr. William Joe 

Thank you for providing us with the reviewers' comments. We thank also the reviewers for their careful review and constructive comments, which have helped to improve the manuscript. We have made changes to the manuscript based on reviewers' suggestions and describe these changes in the below paragraphs. We hope that you will find our responses acceptable and we look forward to your decision.

Reviewer #1: The authors have taken care of all the comments and suggestions. The manuscript seems improved and revised in logical manner. So, there is not more suggestions other than about the title of the paper.

Response: We think that the phrase ‘health belief theory framework’ is superfluous in the title, so we removed it to reduce the verbiage and make the title eye catching. 

Reviewer #2 Please refer to the attached manuscript marked up with my feedback

1. This hyphen should be removed. (Evidence-based)

Response: we have revised Evidence-based as Evidence based (line 94)

2. No need to capitalize (Structural Equation Model)

Response: we have revised Structural Equation Model as structural equation model (line 109)

3. According to the HBT…..

Response: we have removed article ‘the’ in the phrase “According to the HBT” (line 114)

4. Would be interesting to add at least a sentence describing why HBT has been applied disproportionately to Western countries.

Response: We don’t think there is any specific reason that HBT has been only applied in Western Countries…Indian studies have not used theory driven approaches as they may not have been aware of such a thing. Most studies in developing countries are conducted by doctors or folks in the medical field and they don’t learn health theories. Still we have added a text that we think may partly be the reason why HBT might been applied mostly in western countries. The revised text reads as “HBT has, however, been developed to understand predictors of disease prevention behaviors in the United States and is mostly used as a framework for explaining vaccination in Western countries.” (line 122)

5. Describe why girls specifically?

Response: We have provided the reasons why only girls were selected. The added text reads as “This study is a secondary analysis of data collected for a project that examined factors affecting parental intention to accept HPV vaccine for the daughters in Mysore, India [33,34]. Assessing factors correlated with childhood vaccination was a secondary objective of the project. The Indian government has approved HPV vaccination only for adolescent girls [35,36], the parent study; thus, targeted parents who had at least one adolescent daughter during the study period.” (line 140-144).

6. Replace living with reside 

Response: We have replaced the term living with residing (line 134)

7. Was there a question about the immunization status of the parents as children?

Response: there was no question about the immunization status of the parents included in the questionnaire 

8. Would briefly restate that these categories were defined consistently with the WHO criteria

Response: we have reiterated that the full vaccination definition is based on WHO

The revised text reads as “‘Full vaccination’ described families with children that had received one dose of BCG, three doses of DPT, three doses of OPV and one dose of MCV as recommended by WHO.” (Line 219-221).

9. Please give rationale for choosing the age cutoff at 35 years

Response: Age was initially collected as a continuous variable. However, we assumed a better fit of the SEM model when age included as a categorical/binary variable. In addition, the mean age of the study participants was 38.3 with a standard deviation of 6.6.

10. This seems to be a rushed but potentially important point. The authors imply that the size of this group was small and so would be the impact of their responses on the analysis. This is an overreach (even if it is intuitive) because the impact of their responses was not evaluated.

Response: We thank the reviewer for picking this. In addition to the size of the response, issues related to the model fit and interpretation of the results were reasons for the exclusion of the ‘not applicable’ category of some items used to collect information on parental attitudes about childhood vaccination. However, we acknowledge that removing this category may have affected the coefficient estimated in the Final SEM model. Thus, we have discussed this as a limitation. It reads as follows “Moreover, treating the ‘not applicable’ response to some items used to assess parental attitudes about childhood vaccination as missing data might have biased the estimated coefficients in the SEM model. However, the proportion of the study participants who responded as ‘not applicable’ were ≤1% to some items that examined parental attitudes about childhood vaccination and even none for some of them. As a result, the effect of the ‘not applicable’ response on the coefficient estimates could be very minimal.” (line 433-438)

The text which reads “Similarly, the number of participants who reported ‘not applicable’ or “not sure” were also small (<10), so they were excluded from the analysis” was a typo and have been removed. 

11. Perhaps I'm missing something but how well can one reply to the questionnaire if she/he is illiterate? Further how might literacy have affected response rates or responses? Did the issue of literacy impact the design or interpretation of this study?

Response: We acknowledge that illiterate parents might have gotten support from family members or neighbors. And we have included education status that includes illiteracy as a separate category in the model. In addition, we have discussed the potential impact of the illiterate nature of the parents on their response. The added text in the discussion reads as ‘Illiterate parents might have also received support from family members or friends to respond to some questions leading to information bias.” (Line 430,432).

12. Reiterating here the earlier comment about literacy

Response: We would like to refer the reviewer to our response above.

13. Suggest revising this sentence for clarity

Response: We have revised the text which reads ‘In addition, unlike most many studies in India [12-26], did not use as this study did, a health theory-driven model with robust analytical techniques (SEM).” as “In addition, unlike many studies in India [12-26], the current study analyzed data using a health theory-driven model with robust analytical techniques (SEM).” (line 417,418).

14. And simultaneously controlling for parent age correct?

Response: Yes, we have controlled for the effect age when we analyzed data after stratifying by the number of children. We have revised the text to reflect this as “However, after stratifying data based on the number of children and simultaneously controlling for age, full immunization rates remained similar between parents with one child versus those with two or more. (line 427-429).

15. One would expect the overall reported vaccination rates to have been higher if this was the case correct? “Due to the self-reported nature of the data, parents may have been answered questions differently for reasons of social desirability”

Response, Yes, that is true mostly in the urban population but for the rural population may still have a different perspective due to cultural and religious reasons. However, we thought this has more to do with recall than social desirability bias and removed the text “ Due to the self-reported nature of the data, parents may have been answered questions differently for reasons of social desirability” from the revised MS.

16. Reiterating here the earlier point about the need to explain more clearly why only girls were chosen

Response: We would like to refer the reviewer to our response above.

17. Conclusions: Were any important or interesting outcomes observable or observed with respect to religion? Even if not reliably valuable readers will be looking for some consideration of this topic.

Response: We have added a conclusion remark of the finding about the role of religion on parental perception about barriers to vaccination in the conclusion.

The added text reads as “Area of residence may modify the effect of religion and the number of children in a family on parental perceptions about barriers to childhood vaccination.” (line 454 to 456).

---

## [Decision Letter · Decision Letter 2]

2 Oct 2020

Structural equation modeling to detect correlates of childhood vaccination: a moderated mediation analysis

PONE-D-20-09473R2

Dear Dr. Degarege,

We’re pleased to inform you that your manuscript has been judged scientifically suitable for publication and will be formally accepted for publication once it meets all outstanding technical requirements.

Kind regards,

William Joe

Academic Editor

PLOS ONE

Additional Editor Comments (optional):

Reviewers' comments:

Reviewer's Responses to Questions

**Comments to the Author**

1. If the authors have adequately addressed your comments raised in a previous round of review and you feel that this manuscript is now acceptable for publication, you may indicate that here to bypass the “Comments to the Author” section, enter your conflict of interest statement in the “Confidential to Editor” section, and submit your "Accept" recommendation.

Reviewer #2: All comments have been addressed

2. Is the manuscript technically sound, and do the data support the conclusions?

Reviewer #2: Yes

3. Has the statistical analysis been performed appropriately and rigorously? 

Reviewer #2: Yes

4. Have the authors made all data underlying the findings in their manuscript fully available?

Reviewer #2: Yes

5. Is the manuscript presented in an intelligible fashion and written in standard English?

Reviewer #2: Yes

6. Review Comments to the Author

Reviewer #2: The manuscript is much improved. I have no additional comments.

7. PLOS authors have the option to publish the peer review history of their article (what does this mean?). If published, this will include your full peer review and any attached files.

Reviewer #2: No

---

## [Editor Report · Acceptance letter]

6 Oct 2020

PONE-D-20-09473R2 

Structural equation modeling to detect correlates of childhood vaccination: a moderated mediation analysis 

Dear Dr. Degarege:

I'm pleased to inform you that your manuscript has been deemed suitable for publication in PLOS ONE. Congratulations! Your manuscript is now with our production department. 

Kind regards, 

on behalf of

Dr. William Joe 

Academic Editor

PLOS ONE